# Improving Communications to Increase Nonindustrial Private Forest Landowner (NIPF) Participation in Forest Certification Programs: A Case Study in Arkansas, USA

Elena C. Rubino, Nana Tian * and Matthew H. Pelkki

Arkansas Forest Resources Center, College of Forestry, Agriculture & Natural Resources, The University of Arkansas at Monticello, Monticello, AR 71656, USA; rubino@uamont.edu (E.C.R.); pelkki@uamont.edu (M.H.P.)
* Correspondence: tian@uamont.edu

**Abstract:** Despite the socioeconomic and ecological significance of the 10.4 million acres of forestland owned by nonindustrial private forest (NIPF) landowners across Arkansas (approximately 58% of forestland in the state), only 5% of this land is certified through the American Tree Farm System. As such, understanding how to improve the reach and content of communications to NIPF landowners is vital for expanding certification participation and subsequent improvement of forest management in Arkansas and throughout the southern United States. To explore current and optimal communications to increase NIPF participation, we employed Berlo's source–message–channel–receiver (SMCR, 1960) model to analyze survey data collected from Arkansas NIPF landowners regarding their familiarity with, interest in, and information sources for forest certification programs. Our results indicate that NIPF landowners have a relatively low familiarity with certification programs and a low interest in adopting a certification program regardless of personal involvement throughout the certification process, the transparency of on-sight inspections to the public, and the requirements of forest management plans. However, positive correlations were found between self-reported familiarity with certification programs and the perceived usefulness of various information sources, indicating that communications to NIPF landowners have the ability to be influential. Furthermore, the results showed that the greatest perceived benefits to landowners were improved timber growth and health, better management actions, and environmentally-friendly timber harvesting, whereas the most concerning perceived drawbacks were increased record-keeping and paperwork as well as management costs. These findings will offer actionable insights into future messaging campaigns and provide directions for new approaches of reaching NIPF landowners to increase their participation in forest certification.

**Keywords:** Berlo; messaging; nonindustrial private forest (NIPF) landowners; certification program enrollment; source–message–channel–receiver (SMCR); private forests; communication

## 1. Introduction

Forest certification is a voluntary, market-based approach to recognize and encourage sustainable forest management; approximately 430 million hectares (1.06 billion acres) of forest lands are certified globally [1]. Between Canada and the United States (U.S.), North America contains substantial forest lands, although the distribution of certified forests across the landscape is uneven due to landownership differences between the two countries. Whereas most of Canada's forests are public lands administered by provincial governments (resulting in economies of scale for efficient certification), 58% of forestlands in the U.S. are smaller, privately-owned parcels [1]. More so, 38% of U.S. forestlands are 'family forests' (i.e., owned by families, individuals, trusts, and estates), resulting in unique ownership objectives and management practices [2], including certification enrollment. Therefore, private landowners' management decisions play an important role in achieving the goal of economic, social, and ecological sustainability of forestlands.

Currently, existing forest certification programs in Arkansas mainly include the Forest Stewardship Council (FSC), the American Tree Farm System (ATFS), and the Sustainable Forestry Initiative (SFI). For NIPF landowners, ATFS is the primary certification program in Arkansas and about 5% of NIPF land was certified under this program (approximately 487,826 acres). Comparatively, 1,629,730 acres were certified under FSC and 928,680 acres were certified under SFI [3]. Notably, the FSC and SFI programs are primarily for industrial landowners, and they provide landowners with financial incentives and social recognition for promoting sustainable forest management to achieve economic, environmental, and social needs of society. Several studies have explored nonindustrial private forest (NIPF) landowners' attitudes towards and perceptions of forest certification programs in an effort to promote future enrollment in the United States. For example, research has found a range of support for private forestland certification [4–6], as well as a multitude of drivers for certification program participation, including passion for environmental stewardship [7], education [4], age, and management activity level [8]. Furthermore, research investigating barriers to certification for NIPF landowners has consistently found that certification costs [3,5,9–12] and strict program requirements (e.g., required management plans) [9–11] are significant hurdles for landowners. Of note, Newsom et al. [13] also documented NIPF landowners' lack of knowledge regarding forest certification programs, which serves as an additional obstacle to enrollment. In doing so, they hinted at the importance of communications (specifically with forestry professionals) and outreach in generating support for and enrollment in certification programs [4,13]. Despite this research, there remains a dearth of current information on NIPF landowners' familiarity with and interest in forest certification programs under different program requirements. Furthermore, given the revealed significance of communications on program enrollment [4,13], research that expands on how messaging can promote enrollment would be helpful in increasing enrollment rates.

As such, the major objective of this study was to address this gap in communications-driven certification program enrollment research, where we used a survey to document NIPF landowners' familiarity with, interest in, and information sources for forest certification programs. We then employed Berlo's source–message–channel–receiver (SMCR 1960) model to understand how our survey results can be strategically utilized to aid future efforts to increase certification program enrollment among NIPF landowners. Our findings will offer actionable insights into future messaging campaigns and provide directions for new approaches of reaching NIPF landowners to increase their participation in forest certification.

## 2. Methods

### 2.1. Study Area

We focused this study on NIPF landowners in Arkansas due to the state's high dependence on the forest economy, where approximately 5% of Arkansas' GDP is contributed by the forest economy (the highest rate of any southern state, Pelkki and Sherman [14]). Over half (accounting for 10.4 million acres) of Arkansas' forestland is owned by 345,000 NIPF landowners, yet only 5% is certified across three accessible certification programs in the state [3]. These three programs are the American Tree Farm System (ATFS, certified 487,826 acres), the Forest Stewardship Council (FSC, certified 1,629,730 acres), and the Sustainable Forestry Initiative (SFI, certified 928,680 acres). The certification scheme standards and processes differ among the three programs, but they all aim at improving and enhancing the sustainable management of forests. The currently low certification rates represent a powerful opportunity to grow certification participation among NIPF landowners that would simultaneously ensure private forests are managed sustainably and allow NIPF landowners to tap into markets that are increasingly requiring certificated wood fiber [3].

### 2.2. The SMCR Model

Berlo's [15] sender-message-channel-receiver (SMCR) model of the communication process was originally designed to improve technical communication [16], but it has since

been employed in a range of applications, including communications in tourism [17], nursing [16], and education [18]. Within the model, the source, or sender of information, encodes a message that needs to be communicated. They send this message through a channel, so that it may reach the receiver (if successful), who decodes (i.e., interprets) the message [17,19]. Several factors affect each component. Both the source and receiver are influenced by their communication skills, attitudes towards the audience, knowledge of the message content, and social background and culture. The source is responsible for encoding the message appropriately, which includes its structure, content, treatment, and code, and sending it via an appropriate channel (i.e., through visual, auditory, or other sense channels) [19].

Each component of the model can be a point of communication failure, which provides opportunities for analyses of communications. For example, a source can alienate a receiver using an inappropriate word choice, entirely misidentify a receiver's interests, or use the wrong channel to send information [19]. As the basis of this study, we recognize there is a communication failure resulting in low forest certification program enrollment among NIPF landowners. We applied Berlo's [15] SMCR model (Figure 1) to analyze where communication breakdowns are occurring and suggest improvements to maximize communication success that consider source, message, channel, and receiver attributes.

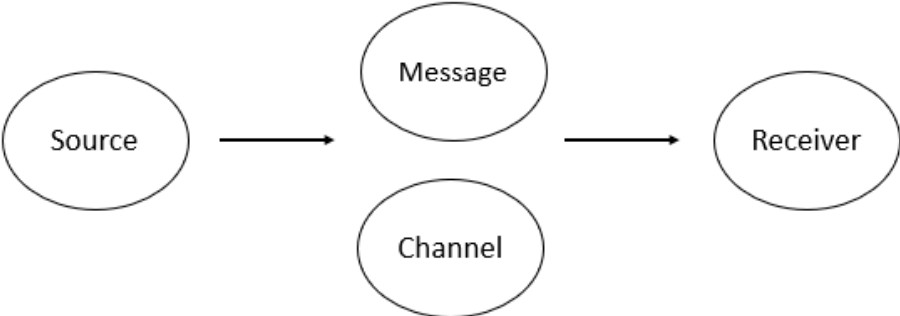

**Figure 1.** Berlo's (1960) SMCR model, adapted from Kuznar and Yager (2020).

*2.3. Data Collection and Analysis*

During the winter of 2020, the University of Arkansas at Monticello conducted a combination of mail and online (using the software of Qualtrics) [20] state-wide surveys of NIPF landowners in Arkansas. Landowners who own at least 10 acres of forestland were randomly selected to receive the survey and the mailing address database was purchased from Dynata Inc. Before implementing the survey, the questionnaire was reviewed and approved by the University of Arkansas at Monticello's Institutional Review Board (IRB No. FNRf-01). A total of 4000 mailings were sent out in the October 2020. We removed 298 questionnaires as ineligible (e.g., undeliverable addresses, death), resulting in a total of 3702 eligible mailings. We received 562 completed surveys from landowners, resulting in a response rate of 15.2%.

The questionnaire consisted primarily of Likert-scale items [20]. For example, participants were asked to rate their familiarity with the concept of forest certification and the most commonly used certification programs in Arkansas (i.e., Forest Stewardship Council (FSC), American Tree Farm System (ATFS), and Sustainable Forestry Initiative (SFI)) on a scale of 1 (not at all familiar) to 5 (extremely familiar). Questions also included landowners' perspectives for possible perceived benefits and drawbacks associated with forest certification (1 = strongly disagree, 5 = strongly agree), as well as their willingness (1 = very unlikely, 5 = very likely) to participate in a certification program under various requirements. On average, it took landowners about 20 min to complete this survey.

Descriptive statistics, including mean value and percentage for the Likert-scale items, as well as sample size, were reported. The analysis of variance (ANOVA) was employed for testing the differences in landowners' willingness to participate in a forest certification program under different program requirements. Chi-square tests can be used for testing



the correlation among categorical/ordinal variables [21]; therefore, chi-square tests were utilized to test the correlation between landowners' familiarity with forest certification and their perspectives on the usefulness of some main information sources. All data were analyzed using the SAS 9.4 program. To be specific, the ANOVA analysis was completed using 'proc anova' and Chi-square tests was carried out using 'proc freq'.

## 3. Results

A total of 562 NIPF landowners responded to the survey, where the majority (69.93%, $n$ = 393) were male and the average age was 61 years old. Nearly half (47.7%, $n$ = 268) of respondents reported having at least an associate degree, while 25.3% ($n$ = 142) indicated having some college education, and 27.0% ($n$ = 152) indicated having a high school-level education or less. Household income levels ranged widely, where 40.6% ($n$ = 228) of respondents reported income between USD 50,000 and USD 100,000, 35.2% ($n$ = 198) between USD 20,000 and USD 49,999, and 7.4% ($n$ = 136) with household income less than USD 20,000.

The mean size of forestland owned by respondents was 68.6 acres, with 17.0% of the respondents reporting ownership of at least 100 acres of forestland. The average length of land tenure was 34 years and the majority of respondents purchased (82.1%, $n$ = 444), rather than inherited (17.9%, $n$ = 97), their lands. Most respondents (54.0%, $n$ = 295) had never harvested their land, 24.0% ($n$ = 131) had harvested over 10 years ago, and 22.0% had harvested within the past 10 years. Relatedly, 63.2% ($n$ = 347) of respondents indicated they do not intend to harvest within the next five years, 12.9% ($n$ = 71) do intend to harvest in the next five years, and 23.9% ($n$ = 131) are unsure.

Respondents were largely not familiar with forest certification programs and systems generally, nor were they familiar with specific options available in Arkansas (Table 1). Specifically, average familiarity with the concept of forest certification in general was highest (1.58) compared to the average familiarity with specific certification programs: FSC (1.29), ATFS (1.34), and SFI (1.29). Whereas approximately 84% of respondents were consistently not at all familiar with specific programs, 68.7% of respondents indicated that they were not at all familiar with forest certification and 14.8% indicated they were slightly familiar with the concept (compared to approximately 7% for specific programs).

**Table 1.** Landowners' familiarity with forest certification programs/systems.

| | | Percentage of Response (%) | | | | | |
|---|---|---|---|---|---|---|---|
| | **Mean** | **Not at All-Familiar** | **Slightly Familiar** | **Somewhat Familiar** | **Moderately Familiar** | **Extremely Familiar** | **$n$** |
| Forest certification concept (in general) | 1.58 | 68.7 | 14.8 | 9.5 | 4.2 | 2.9 | 549 |
| Forest Stewardship Council (FSC) | 1.29 | 84.7 | 7.4 | 3.9 | 2.8 | 1.3 | 541 |
| American Tree Farm System (ATFS) | 1.34 | 83.7 | 6.3 | 5.2 | 2.0 | 2.8 | 539 |
| Sustainable Forestry Initiative (SFI) | 1.29 | 84.6 | 8.0 | 3.0 | 2.6 | 1.9 | 540 |

On average, respondents at least somewhat agreed that forest certification generated perceived benefits in terms of increased timber growth and health (3.40), environmentally-friendly timber harvesting (3.10), and better management practices (3.32) (Table 2). However, respondents tended to only slightly agree that the remaining potential benefits are realized, as average agreement was below 3.0. Notably, nearly a third of respondents did not agree at all that forest certification expanded markets for harvested forest products, resulted in price premiums for harvested forest products, nor generated public recognition for practicing good forestry.

**Table 2.** Landowners' agreement regarding perceived benefits associated with forest certification.

| | | Percentage of Response (%) | | | | | |
|---|---|---|---|---|---|---|---|
| | **Mean** | **Not at All Agree** | **Slightly Agree** | **Somewhat Agree** | **Moderately Agree** | **Extremely Agree** | ***n*** |
| Increased timber growth/health | 3.40 | 17.2 | 8.1 | 20.4 | 25.5 | 28.7 | 494 |
| Expanded product markets | 2.67 | 32.2 | 12.7 | 23.0 | 19.5 | 12.5 | 487 |
| Product price premiums | 2.78 | 30.4 | 12.7 | 23.3 | 16.1 | 17.6 | 490 |
| Public recognition for good practices | 2.63 | 32.4 | 17.5 | 20.0 | 14.7 | 15.5 | 491 |
| Environmentally-friendly harvests | 3.10 | 21.7 | 15.0 | 18.7 | 20.5 | 24.1 | 493 |
| Improved management practices | 3.32 | 18.1 | 10.2 | 21.3 | 22.6 | 27.8 | 492 |

Similarly, on average, respondents slightly agreed with all identified perceived drawbacks of certification. Of note, increased costs of forest management (2.84) and increased record-keeping and paperwork (2.89) garnered the most agreement as perceived drawbacks, on average (Table 3). Approximately 30% of respondents did not agree at all that onsite inspections required forest management plans, and constraints on types of harvesting practices were drawbacks of certification.

**Table 3.** Landowners' agreement regarding perceived drawbacks associated with forest certification.

| | | Percentage of Response (%) | | | | | |
|---|---|---|---|---|---|---|---|
| | **Mean** | **Not at All Agree** | **Slightly Agree** | **Somewhat Agree** | **Moderately Agree** | **Extremely Agree** | ***n*** |
| Increased management costs | 2.84 | 23.7 | 16.2 | 26.2 | 19.9 | 14.0 | 493 |
| Increased record-keeping/paperwork | 2.89 | 23.8 | 14.7 | 25.9 | 19.6 | 16.1 | 491 |
| On-site inspections | 2.60 | 30.7 | 17.6 | 25.4 | 13.3 | 12.9 | 488 |
| Required forest management plans | 2.63 | 29.2 | 18.2 | 26.3 | 13.3 | 13.0 | 483 |
| Constraints on types of harvesting practices | 2.65 | 30.4 | 16.1 | 28.1 | 13.6 | 11.8 | 484 |

Over half of respondents (54.2%, *n* = 291) were not at all interested in having their forestlands certified, with 22.0% (*n* = 118) slightly interested, 12.1% (*n* = 65) somewhat interested, and 11.7% (*n* = 63) moderately or extremely interested. A more specific breakdown of respondent willingness to participate in a forest certification program under different program requirements corroborates this general finding. In all instances, regardless of program requirements, respondents indicated that they were unlikely to participate (Table 4). A few program requirements garnered near-neutral willingness to participate, including educational institution certifying organizations, landowners not being responsible for paying certifying costs, results of on-site inspections not being available to the public, not requiring forest management plans, providing landowners with the ability to choose any logger, and landowners receiving higher prices and preferences from forest product mills (Table 4). Although, we note that these requirements still fall within the "unlikely" range. Program requirements, such as certification through a government organization, processes where landowners were uninvolved and had to pay all the costs to certify, where the results of on-site inspections were made fully available to the public, where forest management plans were required, and where there were requirements to notify the certifying organization of intent to harvest timber, were all considered, on average, to be the most unlikely to garner participation among respondents (Table 4).

**Table 4.** Landowners' willingness to participate in a forest certification program under different program requirements.

| | | Percent of Response (%) | | | | | |
|---|---|---|---|---|---|---|---|
| | Mean | Very Unlikely | Unlikely | Neutral | Likely | Very Likely | n |
| Would you participate if the certifying organization was: | | | | | | | |
| A government organization | 1.88 [a] | 53.8 | 19.4 | 15.6 | 7.9 | 3.4 | 506 |
| a forest products industry association | 1.91 [a] | 52.0 | 39.6 | 17.9 | 9.5 | 2.2 | 504 |
| a forest landowner association | 2.36 [b] | 39.6 | 14.5 | 22.2 | 18.0 | 5.7 | 505 |
| an educational institution | 2.42 [b] | 38.0 | 14.7 | 22.6 | 17.0 | 7.7 | 505 |
| an organization not affiliated with any particular association or group | 2.14 [c] | 44.5 | 16.4 | 24.1 | 10.5 | 4.5 | 506 |
| Would you participate if you were: | | | | | | | |
| required to be involved throughout the process of certifying your forest | 2.28 [a] | 41.8 | 17.8 | 19.3 | 12.9 | 8.2 | 214 |
| required to be involved in some part of certification process | 2.35 [a] | 39.4 | 17.5 | 20.4 | 14.5 | 8.2 | 510 |
| not involved in the certification process | 1.85 [b] | 56.2 | 16.6 | 17.0 | 6.3 | 3.9 | 507 |
| Would you participate if you had to pay: | | | | | | | |
| none of the costs to certify your forest | 2.73 [a] | 37.9 | 10.9 | 13.0 | 16.5 | 21.7 | 515 |
| some of the costs to certify your forest | 1.79 [b] | 58.3 | 15.9 | 16.9 | 6.7 | 2.2 | 508 |
| all of the costs to certify your forest | 1.29 [c] | 81.9 | 10.8 | 4.7 | 2.0 | 0.6 | 507 |
| Would you participate if the results of on-site inspections were: | | | | | | | |
| made fully available to the public | 1.73 [a] | 62.8 | 15.2 | 12.1 | 6.3 | 3.6 | 506 |
| made available to the public only in summary form | 1.97 [b] | 50.8 | 18.3 | 16.7 | 11.0 | 3.2 | 502 |
| not made available to the public | 2.52 [c] | 40.7 | 10.5 | 17.8 | 18.4 | 12.6 | 506 |
| Would you participate if a forest management plan was: | | | | | | | |
| required | 1.96 [a] | 55.5 | 15.2 | 12.4 | 11.2 | 5.7 | 508 |
| encouraged but not required | 2.31 [b] | 37.7 | 19.4 | 21.8 | 16.1 | 4.9 | 509 |
| not required | 2.47 [b] | 38.8 | 14.0 | 19.5 | 16.7 | 11.0 | 508 |
| Would you participate if you were: | | | | | | | |
| required to use a professional forester when managing your forest or harvesting timber | 1.82 [a] | 61.8 | 12.6 | 13.0 | 7.3 | 5.3 | 508 |
| not required to use a professional forester when managing your forest or harvesting timber | 2.28 [b] | 41.6 | 14.7 | 23.5 | 14.3 | 5.9 | 510 |
| required to notify the certifying organization of your intent to harvest timber | 1.71 [a] | 63.0 | 15.5 | 12.1 | 6.7 | 2.7 | 511 |
| required to use only loggers who were trained in environmentally-friendly practices | 2.14 [b] | 50.0 | 16.8 | 13.1 | 9.6 | 10.5 | 512 |
| could use any logger you choose | 2.57 [c] | 26.3 | 13.7 | 20.8 | 15.5 | 13.7 | 510 |
| Would you participate if: | | | | | | | |
| you received a higher price for your timber | 2.89 [a] | 31.6 | 8.6 | 19.6 | 21.0 | 19.2 | 510 |
| you received the same price for your timber | 2.24 [b] | 39.3 | 14.5 | 31.4 | 12.6 | 2.2 | 509 |
| forest product mills gave preference to buying timber from certified forests | 2.65 [a] | 34.2 | 8.6 | 26.6 | 19.9 | 10.7 | 512 |
| forest product mills gave no preference to buying timber from certified forests | 2.22 [b] | 38.0 | 14.6 | 36.4 | 9.3 | 1.8 | 508 |

Note: [a,b] and [c] in the Mean column indicate statistically different at the 0.05 significance level.

Over half of all respondents reported each identified source of information about forest management certification as not at all useful, with the exception of discussions with other landowners, although this option was still considered only slightly useful, on average (Table 5). Of note, despite its low mean score, talks with foresters and professionals were considered moderately or very useful among nearly a quarter of respondents (Table 5). There were significant positive correlations between the perceived usefulness

of all information sources and self-reported familiarity with the concept of forest certification (discussion with forestry professional: $\chi^2 = 121.574$, $p < 0.01$; discussion with fellow landowner: $\chi^2 = 65.268$, $p < 0.01$; on site forestry field trip: $\chi^2 = 68.208$, $p < 0.01$; workshop: $\chi^2 = 71.644$, $p < 0.01$; webinar or video conference: $\chi^2 = 54.387$, $p < 0.01$; book: $\chi^2 = 78.481$, $p < 0.01$; newsletter, magazine, or newspaper: $\chi^2 = 53.592$, $p < 0.01$; video tape: $\chi^2 = 42.464$, $p < 0.01$; television or radio segment: $\chi^2 = 36.346$, $p < 0.01$; website: $\chi^2 = 70.698$, $p < 0.01$).

**Table 5.** Landowners' main sources of information about forest management certification.

| | | Percentage of Response (%) | | | | | | $\chi^2$ Test | |
|---|---|---|---|---|---|---|---|---|---|
| | Mean | Not at All Useful | Slightly Useful | Somewhat Useful | Moderately Useful | Very Useful | n | *p*-Value |
| Discussion with forestry professional | 2.20 | 53.5 | 9.5 | 13.4 | 11.1 | 12.5 | 432 | <0.01 |
| Discussion with fellow landowner | 2.27 | 40.0 | 18.2 | 23.2 | 12.0 | 6.6 | 440 | <0.01 |
| On site forestry field trip | 1.73 | 66.9 | 8.6 | 13.4 | 4.4 | 5.4 | 411 | <0.01 |
| Workshop | 1.70 | 68.2 | 8.6 | 13.4 | 4.4 | 5.4 | 409 | <0.01 |
| Webinar or video conference | 1.61 | 71.3 | 8.3 | 12.9 | 3.2 | 4.4 | 411 | <0.01 |
| Book | 2.00 | 53.9 | 12.8 | 18.8 | 8.3 | 6.2 | 421 | <0.01 |
| Newsletter, magazine, or newspaper | 1.94 | 54.5 | 14.6 | 17.9 | 7.9 | 5.0 | 418 | <0.01 |
| Video tape | 1.54 | 75.1 | 8.3 | 8.8 | 3.4 | 4.4 | 410 | <0.01 |
| Television or radio segment | 1.72 | 63.5 | 12.7 | 15.6 | 4.3 | 3.8 | 416 | <0.01 |
| Website | 1.85 | 62.0 | 10.4 | 14.3 | 6.8 | 6.5 | 413 | <0.01 |

## 4. Discussion

Despite a self-reported lack of familiarity with forest certification programs, there was an overall belief among respondents that forest certification is not beneficial with regards to expanding forest product markets, creating price premiums for products, and generating public recognition, and that forest certification was detrimental in that it increases forest management costs, record-keeping, and paperwork. These perceptions are corroborated by the fact that, on average, respondents were unlikely to participate in forest certification programs, regardless of program requirements.

Overall, our results are often consistent with those of other studies. Unfamiliarity of NIPF landowners with forest certification programs has been recorded in Mississippi and Louisiana [5], Tennessee [4], Minnesota [10,11], and throughout the United States [22]. Additionally, NIPF landowners have frequently expressed low economic benefits from certification (e.g., price premiums, expanded markets) [7,10]. Andersen [7] found respondents to have strong environmental values, which supports our results that environmentally-friendly timber harvesting and better management practices are perceived as benefits of certification. However, he concluded that public recognition benefits were a powerful motivator of certification among Washington NIPF landowners, whereas our respondents did not agree with recognition as a benefit. With regards to information sources, our findings that discussions with other landowners and forestry professionals are perceived as the most preferred options is consistent with multiple studies [10,23], although there are some discrepancies in the literature about which of the two are more useful [23,24].

*Applying the SMCR Model to Forest Certification Program Perceptions*

Such unfamiliarity with and negative perceptions of forest certification programs, paired with findings about preferred information sources, provide ample opportunities to improve certification programs' communication campaigns among NIPF landowners. Furthermore, our results demonstrating significant positive correlations between the perceived usefulness of information sources and familiarity with forest certification indicate that NIPF landowners would find value in improved communications. Indeed, Auld et al. [25]

noted that forest certification outreach efforts have historically largely targeted industrial landowners, public land ownerships, and wood product manufacturers. By situating our findings within Berlo's [15] SMCR model, we can provide insights into the types of communication strategies that will be most effective in resonating with NIPF landowners in the future (Figure 2).

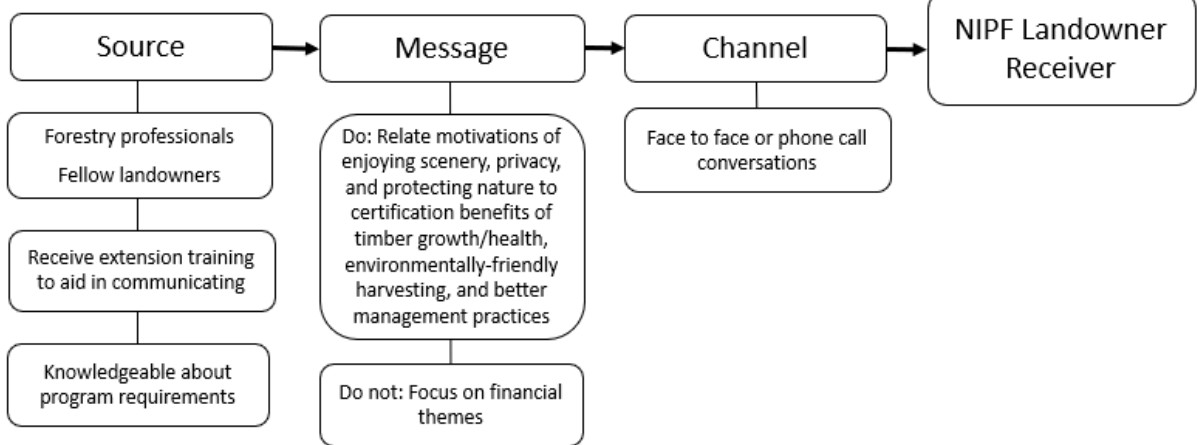

**Figure 2.** Applying Berlo's (1960) SMCR model to NIPF landowner preferences.

For example, landowners and forestry professionals were viewed as at least slightly useful sources of information [10,23], thus we recommend using these individuals (or representative organizations) as sources of communications about forest certification (Figure 2). Within the SMCR model, the efficacy of a source is influenced by characteristics, such as their social system, culture, communication skills, and knowledge [19]. Using fellow landowners and forestry professionals as sources will help ensure that the source and receiver (NIPF landowners) are socio-culturally aligned. Furthermore, providing trainings on extension communications, program requirements, and implementation will improve their communication skills and knowledge base (Figure 2), ultimately making these sources more valuable to NIPF landowners.

To aid in choosing strategic messaging that resonates with NIPF landowners, our results indicate that respondents already perceive increased timber growth and health, environmentally-friendly timber harvesting, and better management practices as benefits of forest certification. These findings can be used in conjunction with findings from Tian and Pelkki [3], who revealed that this same population of NIPF landowners identified the top three most important reasons for owning forestland as to enjoy the forest scenery, for privacy, and to protect nature and biodiversity. Messaging that highlights the relationships between these certification benefits and NIPF landowners' motivations for owning forestland will likely be the most useful to convert NIPF landowners to forest certificatio enrollment.

We saw minimal agreement with expanded markets and price premiums for products as perceived benefits of forest certifications. Furthermore, our results indicated that respondents viewed increased costs of forest management and record keeping in paperwork as substantial perceived drawbacks of certifications. Pairing this with Tian and Pelkki's [3] finding that timber production was the least important reason for NIPF landowners owning forestland, we can infer that using financial themes in messaging will not resonate. This corroborates our findings that respondents indicated they were unlikely to enroll in a certification program, even if it could generate higher prices and preferences from forest product mills. In all, we can conclude that financial motivations are not driving these NIPF landowners' decision-making to enroll in certification programs, and thus this line of messaging should not be the primary tactic.

Although we did not explicitly parse out questions about communication channels, discussions with other landowners and forestry professionals were perceived as more use-

ful, especially as compared to workshops, webinars, and other more traditional mediums (e.g., newsletters, websites, videos). We infer from these disparities that the conversational aspects of these options also makes them more attractive to respondents (in addition to the source of the information). With that in mind, we recommend face-to-face or phone conversations with fellow landowners and forestry professionals (Figure 2), despite that they are likely less cost and time efficient. We do note that this appears to contradict results by Peters [26], who found that nearly 60% of NIPF landowners in North Carolina preferred mailed and online materials, compared to approximately 50% who preferred in-person or online programs. However, these latter options were program-based, and the author did not offer one-on-one, conversational options [26].

## 5. Conclusions

The purpose of our study was to help fill the gap regarding NIPF landowners' familiarity with and interest in forest certification programs under different program requirements. Additionally, we sought to broaden the research available on how strategic messaging can be used to promote certification enrollment. Our results indicate that NIPF landowners have a relatively low familiarity with certification programs and a low interest in adopting a certification program, regardless of personal involvement throughout the certification process, the transparency of on-sight inspections to the public, and the requirements of forest management plans. However, positive correlations between self-reported familiarity with certification programs and the perceived usefulness of various information sources provide evidence that communications to NIPF landowners can be influential. Results indicating that the greatest perceived benefits of certification to landowners were improved timber growth and health, better management actions, and environmentally-friendly timber harvesting offer opportunities for salient messaging. Although there is likely no one-size-fits-all approach to improving forest certification enrollment among NIPF landowners across the United States, using the SMCR model [15], we can be thoughtful and intentional about our communication strategies. Future work can continue to build from these insights to experimentally test message content and framing to optimize how communication campaigns resonate with NIPF landowners. We encourage state forestry agencies, the U.S. Forest Service, and other organizations interested in improving NIPF landowner certification enrollment to reconsider their communication strategies to better reflect the perceptions and needs of landowners. By taking a more strategic approach, such organizations can more effectively shape landowner attitudes towards certification, leading to increased enrollment in the long term.

It is worth noting a few limitations of this study. First, although the response rate of this study was acceptable and on par with other landowner surveys [27], budget limitations prevented us from conducting a nonresponse bias follow-up survey, thus there may be differences between our respondents and individuals within the sampling frame who did not respond. However, we compared the landowner profile between our sample results with the results of the National Woodland Owner Survey in Arkansas [28] and found considerable similarities. For example, landowners in our sample averaged 61 years of age, compared to 67 years of age in the national survey. Second, considering the national and regional differences of social and cultural context among NIPF landowners, our findings may not hold in other places. However, we contend that regardless of location or cultural context, intentional and targeted communication strategies with landowners can be used to improve their knowledge and interest in certification programs, and our results can provide an initial baseline of information for others.

**Author Contributions:** Conceptualization, M.H.P. and N.T.; methodology, N.T.; formal analysis, N.T.; writing—original draft preparation, E.C.R.; writing—review and editing, E.C.R. and N.T.; project administration, N.T.; All authors have read and agreed to the published version of the manuscript.

**Funding:** This research did not receive any specific grant from funding agencies in the public, commercial, or not-for-profit sectors.

**Institutional Review Board Statement:** The study was conducted in accordance with the Declaration of Helsinki, and approved by the Institutional Review Board (or Ethics Committee) of University of Arkansas at Monticello (protocol code IRB No. FNRf-01 and 10 September 2020).

**Informed Consent Statement:** Informed consent was obtained from all subjects involved in the study.

**Data Availability Statement:** Not Applicable.

**Acknowledgments:** We are thankful for the support provided by the College of Forestry, Agriculture & Natural Resources and the Arkansas Center for Forest Business at the University of Arkansas at Monticello in completing this study.

**Conflicts of Interest:** The authors declare no conflict of interest.

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
