# Peer review of "Improving Communications to Increase Nonindustrial Private Forest Landowner (NIPF) Participation in Forest Certification Programs: A Case Study in Arkansas, USA"

_forests, doi:10.3390/f13010086_

Round 1

Reviewer 1 Report

The manuscript is aimed to actual and interesting topic related to communication improvement in the form of increase nonindustrial private forest landowner (NIPF) participation in forest certification programs (case study Arkansas). The paper requires a minor revision and corrections to improve the overall structure and readability. I have following questions and suggestions for the authors:

L2-3: I suggest the authors to change the title in form of Arkansas case study approach!  

L35: The choice of keywords, especially when abbreviations are used, is not exactly the most appropriate! I suggest the authors to include forestry or private forests, certification program, communication, etc.

It would be desirable to include in the Introduction chapter and explain in 3-6 sentences the dominant certification programs (FSC, PEFC, ATFS, SFI) in private and public forests at the USA level!

In introduction chapter, did I miss disposition paragraph?

L73-95: I suggest the authors to move the chapter under the title “The SMCR Model” from Introduction to the Methods chapter (2.1. Study area; 2.2. The SMCR Model; 2.3. Data collection and analysis)!

L133-140: Please list the programs you used to create and check the database and the program you used to conduct statistical tests!

Table 1, 2 and 3 - it is necessary to explain the results in the above tables a little more widely and that the textual explanation accompanies each table separately!

L191: Table 4 is quite large, is there a possibility to show the key things in it or to show it graphically or a less significant part as an appendix!

L231: The chapter title “Applying the SMCR Model to Forest Certification Program Perceptions” within the discussion is more relevant to the research result and would be more appropriate to be in the results as chapter 3.1. Suggestion for authors is to transfer part of the text and figure 2 to the results and leave the rest of the text that is adequate for discussion in the existing chapter!

L262: The names of the authors of the paper (Tian and Pelkki’s) are missing in the sentence before the citation under number 12!

In the discussion part where are your limitations regarding your research?

I would like to see more precise conclusions, evidently matching your purpose (such as, In conclusion…).

Reviewer 2 Report

The revised article follows up on the results of a survey published in the journal Forest Policy and Economics (10.1016/j.forpol.2021.102552). The disproportionate similarity of the two articles was not found.

The article provides interesting information related to the improving communications to increase nonindustrial private forest landowner participation in forest certification programs. Statistical methods are used to achieve research results - without comments.

I have the following comments on the article:

I miss the indication of the specific statistical tools used (lines 133-140). Did you use eg. the SPSS program? What version of the program?

The Conclusion summarizes the achieved results. However, given the focus of the topic of the article, I lack at least brief information on the potential to reflect the results into regional or national policy / certification schemes / strategy documents etc.

I have no significant negative comments on the article.

Reviewer 3 Report

Dear Authors,

Thanks for your parer on perceptionf of forest certification and possible communication in this regars.

Here few suggestions from rewiew:

L35 - Please consider to select keywords better connected to 
the hypothesis of this research - forest certification, non-
industrial forest owners, communication

L122 - Reasons of reduction from total of 3,702 eligible 
mailings to 562 usable surveys should be explained. The fact 
that 85% of eligible responses were unusable deserves an 
interpretation (this is probably the main result of the survey).

L292 Conclusions should highight key findings.
